# Towards Innovation Performance of the Hospitality and Tourism Industry: Interplay among Business Ethics Diffusion, Service Innovation, and Knowledge-Sharing

Hongzhou Yuan [1], Ming Su [2,*], Justyna Zywiolek [3], Joanna Rosak-Szyrocka [3], Asad Javed [4] and Zahid Yousaf [5,*]

1 School of Management, Guangzhou College of Commerce, Guangzhou 511363, China
2 School of Economics and Management, Henan Institute of Science and Technology, Xinxiang 453003, China
3 Department of Production Engineering and Safety, Faculty of Management, Czestochowa University of Technology, 42-200 Czestochowa, Poland
4 Department of Management Sciences, Hazara University, Mansehra 23100, Pakistan
5 Higher Education Department, Government College of Management Sciences, Mansehra 23100, Pakistan
* Correspondence: ming020100803@163.com (M.S.); muhammadzahid.yusuf@gmail.com (Z.Y.);
  Tel.: +92-32-1980-4474 (Z.Y.)

**Abstract:** This paper examines the direct relationship between business ethics diffusion and innovation performance. This study also investigates the mediating role of service innovation and the moderating role of knowledge-sharing between business ethics diffusion and innovation performance. This is quantitative research, and cross-sectional data were collected from hotels and tourism firms through adapted scales. The results show that business ethics diffusion and innovation performance are directly linked. Service innovation acts as a mediator between business ethics diffusion and innovation performance. Findings also demonstrate that knowledge-sharing moderates significantly between business ethics diffusion and innovation performance. Service innovation performs a crucial role in enhancing the innovation abilities of the hospitality and tourism industry. The current study shows that knowledge-sharing enhances the effects of business ethics diffusion on innovation performance. The theoretical model enlightens the critical role of business ethics diffusion in promoting a high level of service innovation among individuals to boost innovation performance for the hospitality and tourism industry.

**Keywords:** business ethics diffusion; innovation performance; service innovation; knowledge-sharing; hospitality and tourism industry

## 1. Introduction

In modern-day business, many organizations are facing the problem of unethical behavior and bribery. Various researches presents that the diffusion of unethical behavior among stakeholders results in a self-reinforcing cycle [1]. Many scholars believe that promotion, training, and selection can convey, transmit, and replicate moral traditions, and there is chance for sustainable diffusive ethics to be beneficial for the attainment of innovation performance in a firm [2]. Ethics diffusion provides opportunities for the adaptation of advanced innovative patterns to solve difficulties in a better and faster way [3]. Therefore, in our study, we tested how ethics diffusion efficiently operates, and service innovation benefits the attainment of innovation performance. Business ethics diffusion is entirely dependent on the people who emphasize innovation performance activities [4]. The topic of innovation performance receives much attention from various researchers and practitioners, particularly in the hospitality and tourism industry. Due to ever-increasing pressure from competitors in local, as well as foreign, firms, hospitality and tourism firms reflect on innovation practices as a critical factor in success; therefore, they are grappling

with the transformation from being imitators to innovators [5]. Therefore, these firms continuously search for novel, specific, and effective ways to explore and enhance the innovation capacities of their firms [6]. Ethics diffusion helps to protect and support values, for instance, respect or trust, etc., which results in making employees feel empowered and safe in a variety of situations [7].

In the current study, we explored whether business ethics diffusion, knowledge-sharing, and the mediating role of service innovation are vital potential factors that positively influence the innovation performance of the hospitality and tourism industry. However, empirical evidence about how the mechanism of service innovation mediates the association between business ethics diffusion and innovation performance is not probed in depth, especially with the mediating role of knowledge-sharing. Thus, in our study, we spotlight the knowledge-sharing moderator and service innovation as a mediator in the relationship between business-ethics diffusion and the innovation performance of tourism firms. This research model is interesting, unique, and important for the following reasons. First, business-ethics diffusion and service innovation are considered key antecedents of successful innovation performance. Empirical facts in the existing literature on the association between business-ethics diffusion, service innovation, and innovation performance are sparse or limited primarily in the perspective of tourism firms. Furthermore, there is also a need to recognize antecedents, such as service innovation, which acts as a bridge in business ethics diffusion and innovation performance links. Thus, investigating how business ethics diffusion and service innovation stimulate innovation performance in the hospitality and tourism industry is indispensable. Secondly, employee service innovation practices and processes help to turn the entity's expertise, ideas, and knowledge into the firm's capital and knowledge, which are precursors to increasing the innovation performance of hospitality and tourism firms. However, service innovation is dependent on the willingness of firms to change their traditional patterns and innovate to increase their innovation capabilities and performance. To address and alleviate these issues, hospitality and tourism firms can improve service innovation activities among firms to foster their innovation performance through knowledge-sharing. Accordingly, examining the mediating role of service innovation between business ethics diffusion and the innovation performance relationship is needed to enhance understanding and improve the innovation performance of hospitality and tourism firms. Thirdly, knowledge-sharing has a crucial role in establishing positive surroundings that can encourage employees to collect, implement, and share their knowledge with one another in a firm. Service innovation through different methods and ideas supports the effectiveness of the association between business ethics diffusion and innovation performance. Consequently, studying the knowledge-sharing potential playing a moderating role is significant in enhancing our understanding of the association between business ethics diffusion and innovation performance. Given such background, to fill theoretical and also practical gaps, our study designed a theoretical model to examine the impact of business ethics diffusion on the innovation performance of hospitality and tourism firms through service innovation (mediating role) and knowledge-sharing (moderating role).

The population of the current study is all the managers, CEOs, and owners of hospitality and tourism firms working in Pakistan. The authors selected Pakistani firms as it is a developing nation where tourism has recently been developed to promote economic growth and improve the quality of life of its citizens. Pakistan is now seeing the fastest growth rates in tourist arrivals in the whole world. Pakistan is host to some of the best tourist resorts in the world, K-2, the world's second-largest mountain, three mountain ranges, different picnics pots, and the world's oldest religious places, such as the Buddhist temple and Taxila ruins, etc. Pakistan is also a charming four-season picnic point for foreigners. Thus, it is essential to study and develop a comprehensive model for enhancing the innovation performance of the industry.

The current research used SEM (structural equation-modeling) to investigate the relationship between the antecedents in the proposed research based on data collection

from 570 respondents in tourism firms. The list of the selected tourism firms was obtained from SMEDA (small and medium-enterprises development authority). The tourism firms that were selected play a key role in social and environmental development. A total of 389 such firms were identified and considered for the research. The current research aims to clarify the following study questions.

Q1: Is business ethics diffusion positively linked to innovation performance?

Q2: Does knowledge-sharing moderate business ethics diffusion's effect on innovation performance?

Q3: Does service innovation mediate business ethics diffusion's effects on innovation performance?

For a better understanding of the paper, the paper is divided into different sections. The first section is the Introduction, which explains the background and research gap. Section 2 includes the literature review, and Section 3 contains details about the methods and material for the current research. Section 4 presents data analysis and a discussion of empirical results. Lastly, the discussion and conclusion are given in Section 5.

## 2. Literature Review

Operational definitions of all the variables are given:

### 2.1. Business Ethics Diffusion

Diffusion of business ethics is intrinsic to innovation and acts to influence the firm's survival and develop competencies in the industry [8]. It is defined as a person's or respondent's belief that applying ethics in business results in the creation of an overall positive business environment [9].

### 2.2. Service Innovation

This refers to continuous improvement via innovation in different products [10]. Service innovation involves different interaction channels, various service distribution systems, technological perception of the aforementioned items, etc. [11].

### 2.3. Innovation Performance

Innovation performance refers to the ideas and creativity to improve products, procedures, and processes that may raise the performance and effectiveness of products or services [12].

### 2.4. Knowledge-Sharing

Knowledge-sharing is the practice of exchanging and sharing ideas, perceptions, and information between different teams, groups, people, and the organization [13] that enhances overall performance.

### 2.5. Business Ethics Diffusion and Innovation Performance

The diffusion of business ethics practices is fundamental to the innovation performance of affirm. It is also essential for the advancement of personal capabilities and the survival of business in the current business setting. Ethics diffusion can be understood as positively developing ethical surroundings inside companies [14]. Business ethics act in different ways, including in terms of dogma and law context, etc., which prompt innovative action [15]. A firm or individual inside the firm should perform ethically to cope with the challenges of innovation performance [16]. Business ethics' normative and receptive nature imitates modern norms and principles to be accepted in the current business world and, accordingly, forces them to modify their innovation performance practice [17]. The main focus of business ethics diffusion is not on self-standards, yet it emphasizes the capability of receptiveness to deal with different societal expectations in order to increase innovation performance [18]. Nevertheless, ethical firms are mostly competent and flexible to the customer's needs, which positively affect innovation performance [19]. Ethical firm

management commonly recognizes business ethics as a fundamental need when concentrating on customer interests/needs. Addressing customer interests not only positively influences innovation performance but is also ethically important for the business [20]. Prior literature also shows the positive association between good business ethics diffusion and innovation, which boosts worker performance and satisfies customer needs. Lastly, all reap the benefits of the innovation performance [21]. Business ethics diffusion is observed within an organization through its plans, practices, and activities that eventually attract imitation [22]. This course of action has led to proposing the following hypothesis.

**Hypothesis 1 (H1).** *Business ethics diffusion positively influences innovation performance.*

### 2.6. Service Innovation as Mediator

The literature on business has mostly focused on the significance of participative practices in the diffusion of business ethics and the improvement of innovation performance [23]. In existing research, researchers emphasized ethics diffusion strategies and contended that competitive benefits arise from innovation performance [24]. Therefore, in the current research, we examine the mediating role of service innovation in the link between business-ethics diffusion and innovation performance. Service innovation refers to the implementation of successful creative ideas in the firm [25]. It acts as an intrinsic means to adapt business ethics in the dynamic environment of a company, which increases innovation performance. Thus, firms are required to design new, innovative ideas to offer service innovation [26]. Business ethics diffusion supports firms in performing ethically according to the benefits of society and itself by bringing service innovation via various products and processes that enhance innovation performance [27]. Ethics diffusion does not mean diversification from existing business practices [28]. However, service innovation improves the skills, capacity, and competencies of a firm's employees, which leads to improvements in innovation performance through new methods and technologies [29]. Service innovation is the junction of the production and the consumption processes, which are constantly involved in reforming the products and methods featuring high-level intangibility [30]. Innovation is a critical factor if a firm wants to grow, expand, and continue as a competitor in an emerging market [31]. The results of the current research show an unelectable mediating role of service innovation in the association between business-ethics diffusion and innovation performance. From the above argument, we formulate the second hypothesis.

**Hypothesis 2 (H2).** *Service innovation plays a mediating role between business-ethics diffusion and innovation performance.*

### 2.7. Knowledge-Sharingas a Moderator

Knowledge-sharing is primarily concerned with the understanding and transfer of experience, job processes, and values among fellow employees. It is a critical and intangible asset of a firm that leads to a firm's competitiveness and sustainability [32]. Practically, business ethics diffusion is a process of knowledge-sharing and improving ethical behavior within society, which enhances the innovation performance of a firm [33]. The association between business-ethics diffusion and innovation performance is positively moderated by knowledge sharing. People spread/increase knowledge via sharing knowledge, which is the most important dilemma in the management of knowledge [34]. Knowledge management frequently requires business ethics diffusion that supports the exchange of ideas and trust among employees, which enhances the innovation performance of the firm [35]. In ethical firms, workers share knowledge and concepts with fellow members, and via teamwork, all members should give their views [36]. Consequently, one idea or concept, through interaction, modification, and common consensus, transforms into a different innovative idea, and the firm's knowledge grows in a continuous cycle that helps boost innovation performance [37]. Previous literature also supports that knowledge-sharing moderate be-

tween ethical business units, which effectively augments a firm's innovative capability [38]. Business-ethics diffusion and high innovation performance in the firm also revealed that most often, when team members share their ideas and knowledge, a high level of innovation performance is achieved [39]. Thus, when a high level of knowledge-sharing occurs in an organization, it improves innovation performance [40]. This shows that knowledge-sharing moderates between business-ethics diffusion and innovation performance links.

**Hypothesis 3 (H3).** *The link between business-ethics diffusion and innovation performance is moderated by knowledge sharing.*

Figure 1 shows theoretical framework of this research.

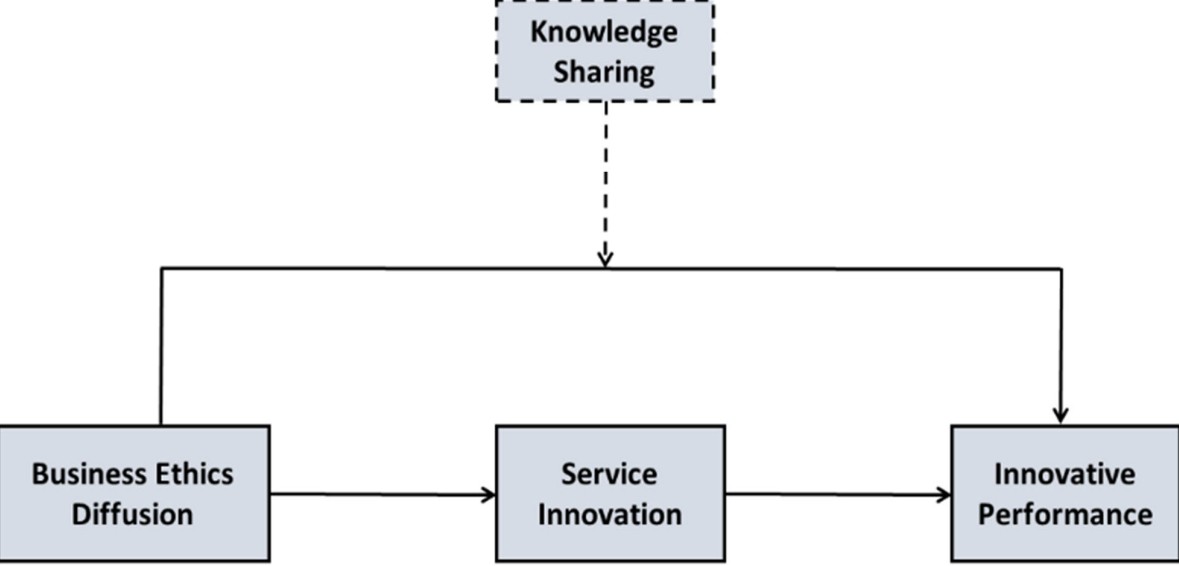

**Figure 1.** Theoretical Framework.

### 3. Methodology

The main purpose of this study was to evaluate the impact of BED on IP and assess the mediating role of service innovation between them and analyze the extent of the influence of the moderating role of knowledge-sharing between BED and IP links. To achieve the objective of the study, we chose participants from the hospitality and tourism industry in Pakistan and selected those firms that represent a noteworthy number of jobs while also having positive societal impacts. For sampling, we targeted respondents such as executives, CEO, and senior managers who were well-known enough with the ethics of the business and other aspects of the firm to respond to the questionnaire. A total of 570 questionnaires were distributed among respondents in hard form, out of which 436 responses were received back from respondents, in which only 389 responses were complete and further used for the analysis. This makes the return rate 68.24%. The list of the selected hospitality and tourism firms was acquired from SMEDA (small–medium enterprises development authority). Hospitality and tourism industry firms reflecting 389 valid responses play a key role in social and environmental development. These hospitality and tourism firms are prominent representatives of the service industry.

The study selected only hotels with a good reputation that were not involved in any unethical practices regarding employee selection and payments to their employees. Furthermore, prior consent was taken from all the respondents. Respondents were not forced to complete the questionnaire, and they could quit the survey at any time. They were assured that the data they provided would be kept secret and only used for the purpose of this research.

*3.1. Measurements*

This research questionnaire was adapted in line with prior studies and literature on the related ethical attributes. The questionnaire was separated into two sections. The first section contains the demographic variables, for instance, respondents' age, field experience, education, etc. Section 2 includes the study items. The validity and reliability of the questionnaire were established by three academic experts before their distribution. To measure the study items, a 5-point Likert scale was developed in which 1 = strongly disagree and 5 = strongly agree.

*3.2. Business Ethics Diffusion*

Business ethics diffusion is measured through six items that compute the perceived business ethics diffusion level. These items were obtained from Roger, 1962 [41]. This construct measures the participant's belief that implementing business ethics is better than not practicing it. The sample question is 'Diffusion of business ethics enhanced performance than not applying/practicing it'.

*3.3. Knowledge-Sharing*

For the measurement of the knowledge-sharing level, a five-item scale was used, which was adapted from Wu, 2016 [42]. This construct shows the real condition of the employees who consider knowledge-sharing beneficial and valuable. The example item is 'In our firm we consider that sharing of knowledge is beneficial for all'.

*3.4. Service Innovation*

Service innovation is measured through a seven-item scale which was developed by Grawe, Chen, and Daugherty, 2009 [43] and also used by Vuori and Okkonen [44]. This construct measures the variety of the methods or tools used in the firm and how services make customers feel respected and special. The example item is 'In project management service innovation is enthusiastically accepted' or 'We constantly find novel patterns for better servicing our customers'.

*3.5. Innovation Performance*

Innovation performance is measured with a five-item scale adapted from Ibarra, 1993 [45] and also used by Huang and Li, 2009 [46]. This construct measures the level of innovation activities in the firm. The sample item is 'Our firm plan different innovative procedures for product designing'.

*3.6. Control Variables*

We used three control variables in this study, including respondents' age, education, and field experience.

## 4. Analysis

We conducted a pilot study, and the results all show acceptable results. In addition, we pre-tested our questionnaire with three expert's from the industry and three academic researchers, and as per their recommendations, a few changes were incorporated into the final questionnaire. Furthermore, items used in this questionnaire were also adapted from prior studies that had similar criteria and different study areas. We conducted confirmatory factor analysis (CFA) to observe variables of business-ethics diffusion, knowledge-sharing, service innovation, and innovation performance. According to the [47], we confirmed that the used hypothesized empirically investigated model best fits the data. Our four-factor model fit to the data, and three substitute models were rejected. The fit keys, $\chi^2$ = 1045.52, CFI = 0.93, GFI = 0.92, RMSEA = 0.05, TLI = 0.95, and SRMR = 0.07 showed the overall model fitness.

### 4.1. Reliability and Validity

SPSS 23.0 was used to conduct analysis in the current research. Table 1 illustrates the results of the convergent validity, the average value extracted, and Cronbach's alpha [48]. The recommended method of [49] was applied to test the discriminant validity. Table 1 shows that all values confirmed it as composite reliability and AVE were higher than the cut-off points, i.e., CR was greater than 0.70, AVE was greater than 0.50, and CR was greater than the average variance extracted. Cronbach's alpha was greater than 0.70.

**Table 1.** Results of Alpha, Composite Reliability, and Average Variance Extract.

| Variable Description | Fac-L | T-Value | Alpha | CR | AVE |
|---|---|---|---|---|---|
| Business Ethics Diffusion | | | 0.84 | 0.96 | 0.78 |
| BED-1 | 0.83 | 15.54 | | | |
| BED-2 | 0.88 | 15.21 | | | |
| BED-3 | 0.85 | 14.54 | | | |
| BED-4 | 0.76 | 15.74 | | | |
| BED-5 | 0.89 | 15.63 | | | |
| BED-6 | 0.82 | 14.63 | | | |
| Knowledge Sharing | | | 0.82 | 0.98 | 0.76 |
| KS-1 | 0.82 | 15.52 | | | |
| KS-2 | 0.78 | 14.77 | | | |
| KS-3 | 0.84 | 15.63 | | | |
| KS-4 | 0.79 | 14.74 | | | |
| KS-5 | 0.86 | 15.21 | | | |
| Service Innovation | | | 0.88 | 0.94 | 0.72 |
| SI-1 | 0.86 | 15.47 | | | |
| SI-2 | 0.82 | 14.52 | | | |
| SI-3 | 0.76 | 15.74 | | | |
| SI-4 | 0.74 | 15.56 | | | |
| SI-5 | 0.84 | 14.21 | | | |
| SI-6 | 0.82 | 15.48 | | | |
| SI-7 | 0.76 | 14.75 | | | |
| Innovation Performance | | | 0.88 | 0.96 | 0.74 |
| IP-1 | 0.84 | 15.74 | | | |
| IP-2 | 0.76 | 14.53 | | | |
| IP-3 | 0.82 | 15.48 | | | |
| IP-4 | 0.78 | 14.52 | | | |
| IP-5 | 0.74 | 14.64 | | | |

### 4.2. Descriptives

Table 2 shows the outcomes of the descriptive statistics and the correlation of the different research constructs. Results show that all the values are positive and significantly correlated. There is a positive relationship between BED and IP, as depicted by the value of the correlation coefficient, which is (r = 0.26 **, $p < 0.001$). Similarly, the correlation between KS and IP is positive, as shown by the correlation coefficient value (r = 0.34 **, $p < 0.001$). Furthermore, service innovation and IP are also positively correlated with coefficient value (r = 0.28 **, $p < 0.001$). The VIF scores were less than the cut-off values of 10.0, which verified that there was no problem with multi-collinearity.

**Table 2.** Results of Mean, Standard Deviation, and Correlations.

| | Variable | Mean | SD | Alpha | 1 | 2 | 3 | 4 | 5 | 6 | 7 | 8 |
|---|---|---|---|---|---|---|---|---|---|---|---|---|
| 1 | Business Age | 3.02 | 1.07 | 0.82 | 1.00 | | | | | | | |
| 2 | Business Size | 1.22 | 0.42 | 0.85 | 1.72 ** | 1.00 | | | | | | |
| 3 | Respondent Experience | 1.65 | 0.46 | 0.87 | 0.015 | 0.036 | 1.00 | | | | | |
| 4 | Respondent Education | 1.47 | 0.52 | 0.40 | 0.049 | 0.047 | −0.142 | 1.00 | | | | |
| 5 | Business Ethics Diffusion | 3.25 | 0.35 | 0.88 | 0.107 ** | 0.016 | 0.024 | −0.15 | 1.00 | | | |
| 6 | Knowledge Sharing | 3.58 | 0.45 | 0.82 | −0.028 | 0.0820 * | 0.0960 ** | −0.14 | 0.425 ** | 1.00 | | |
| 7 | Service Innovation | 3.59 | 0.78 | 0.85 | 0.016 | −0.02 | −0.011 | 0.092 ** | 0.323 ** | 0.175 ** | 1.00 | |
| 8 | Innovation Performance | 0.28 | 0.44 | 0.81 | 0.022 | 0.001 | −0.03 | −0.03 | 0.263 ** | 0.345 ** | 0.280 ** | 1.00 |

Note: ** = $p$ value significant at 0.000; * = $p$ value significant at 0.05.

### 4.3. Hypothesis Testing

In order to accept or reject the hypotheses based on collected data, we used the structural equation modeling analysis. Table 3 shows that business ethics diffusion is positively and significantly associated with innovation performance (β = 0.24 **, $p < 0.001$). Hence, H1 was supported by the data.

**Table 3.** Business Ethics Diffusion effect on Innovation Performance.

| Model | Hypothesis Description | B | F | T | Sig | Remarks |
|---|---|---|---|---|---|---|
| Model # 01 | Business Ethics Diffusion to Innovation Performance | 0.24 | 16.058 | 0.1245 | 0.000 | Accepted |

Table 4 presents the indirect impact of service innovation between business ethics diffusion and innovation performance. Outcomes indicate that service innovation acts as a mediator (Beta = 0.24, Lower = 0.1875 to Upper = 0.3246). Therefore, H2 was confirmed, and it shows that the BED and IP link is mediated through SI.

**Table 4.** Mediating Effect of Service Innovation between BED and IP.

| Model Detail | Data | Boot | SE | Lower | Upper | Sig |
|---|---|---|---|---|---|---|
| BED→SI→IP | 0.2496 | 0.2845 | 0.45 | 0.1875 | 0.3246 | 0.0000 |

Table 5 shows the moderation results of knowledge-sharing on the direct link between business ethics diffusion and innovation performance. The results indicated that the KS is a positive moderator and plays a significant role in the relationship between BED and IP, i.e., (β = 0.38 **, $p < 0.001$). Hence, H3 was supported by the data.

**Table 5.** Hierarchical regression results for moderating effect of knowledge-sharing.

| | Innovation Performance | | | | | |
| --- | --- | --- | --- | --- | --- | --- |
| **Detail** | **Beta** | **T Value** | **Beta** | **T Value** | **Beta** | **T Value** |
| Step-1 | | | | | | |
| Business age | 0.08 | 0.26 | 0.02 | 1.25 | 0.02 | 0.26 |
| Business size | 0.06 | 0.28 | 0.12 | 0.75 | 0.12 | 0.88 |
| Respondent education | 0.12 | 0.24 | 0.16 | 0.12 | 1.05 | 1.36 |
| Respondent experience | 0.17 | 0.22 | 0.18 | 0.92 | 0.02 | 0.18 |
| Step 2 | | | | | | |
| Business Ethics Diffusion | | | 0.34 * | 7.95 | 0.38 * | 3.52 |
| Knowledge Sharing | | | 0.28 * | 5.73 | 0.37 * | 4.75 |
| Step 3 | | | | | | |
| BED × KS | | | | | 0.38 ** | 2.22 |
| F | | 5.16 ** | | 18.35 * | | 16.25 * |
| R2 | | 0.02 | | 0.28 | | 0.26 |
| R2 | | | | 0.22 | | 0.01 |

Notes * $p < 0.0001$, ** $p < 0.05$ (two-tailed); and results of VIF were below the threshold level.

## 5. Discussion

Our study model is exceptional as it shows the supportive role of business ethics diffusion, service innovation practices, and knowledge-sharing in firms that primarily helps to enhance and attain better innovation performance in the hospitality and tourism industry. Knowledge-sharing is an important component in promoting and attaining innovation performance. According to the findings above, in this study, three hypotheses were developed and tested. H1 presents that business ethics diffusion has a significant and positive influence on innovation performance. It supports decision-makers in promoting perceived levels of employees to successfully adopt innovation performance. H1's outcome demonstrates that the diffusion of business ethics practices is intrinsic to the innovation performance of the firm, the advancement of personal capabilities, and the survival of a business in the advanced business world. Business ethics act in different ways, including in terms of dogma and law context, etc., which prompt innovative action [15]. A firm or individual inside the firm should perform ethically to cope with the challenges of innovation performance [16]. Business ethics' normative and receptive nature imitates modern norms and principles to be accepted in the current business world and, accordingly, forces them to modify their innovation performance practice [17]. The main focus of business ethics diffusion is not on self-standards, yet it emphasizes the capability of receptiveness to deal with different societal expectations in order to increase innovation performance [18]. Nevertheless, ethical firms are mostly competent and flexible to the customer's needs, which positively affect innovation performance [19]. Ethical firm management commonly recognizes business ethics as a fundamental need when concentrating on customer interests/needs. Addressing customer interests not only positively influences innovation performance but is also ethically important for the business [20]. The findings of H1 provide motivation and are consistent with previous research results. Secondly, service innovation assists in the diffusion of business ethics and the attainment of innovation performance.

H2 explores the mediating role of service innovation between business-ethics diffusion and innovation performance. The findings of H2 confirm that business-ethics diffusion is a prerequisite for significantly affecting innovation performance through means of innovation performance. The results support the work of previous research. The literature on business has mostly focused on the significance of participative practices in the diffusion of business ethics and the improvement of innovation performance [23]. In existing research, researchers emphasized ethics diffusion strategies and contended that competitive benefits arise from innovation performance [24]. Therefore, in the current research, we examine the mediating role of service innovation in the link between business-ethics diffusion and innovation performance. Service innovation refers to the implementation of successful

creative ideas in the firm [25]. It acts as an intrinsic means to adapt business ethics in the dynamic environment of a company, which increases innovation performance. Thus, firms are required to design new, innovative ideas to offer service innovation [26]. Business ethics diffusion supports firms in performing ethically according to the benefits of society and itself by bringing service innovation via various products and processes that enhance innovation performance [27]. Ethics diffusion does not mean diversification from existing business practices [28]. However, service innovation improves the skills, capacity, and competencies of a firm's employees, which leads to improvements in innovation performance through new methods and technologies [29]. Service innovation is the junction of the production and the consumption processes, which are constantly involved in reforming the products and methods featuring high-level intangibility [30]. Thirdly, the empirical findings of H3 revealed that knowledge-sharing moderates between business ethics diffusion and innovation performance. The outcomes of H3 are consistent with prior literature that Knowledge-sharing is primarily concerned with the understanding and transfer of experience, job processes, and values among fellow employees. It is a critical and intangible asset of a firm that leads to a firm's competitiveness and sustainability [32]. Practically, business ethics diffusion is a process of knowledge-sharing and improving ethical behavior within society, which enhances the innovation performance of a firm [33]. The association between business-ethics diffusion and innovation performance is positively moderated by knowledge sharing. People spread/increase knowledge via sharing knowledge, which is the most important dilemma in the management of knowledge [34]. Knowledge management frequently requires business ethics diffusion that supports the exchange of ideas and trust among employees, which enhances the innovation performance of the firm [35]. In ethical firms, workers share knowledge and concepts with fellow members, and via teamwork, all members should give their views [36]. Consequently, one idea or concept, through interaction, modification, and common consensus, transforms into a different innovative idea, and the firm's knowledge grows in a continuous cycle that helps boost innovation performance [37]. These outcomes confirmed that innovation performance is achieved through the support of business ethics diffusion of innovative ideas, and knowledge-sharing plays a stimulating role in the whole process.

*5.1. Theoretical Contributions*

This research significantly adds to the literature and knowledge in the following ways. Firstly, in our study, we proposed an empirical model to link business-ethics diffusion, service innovation, and the knowledge-sharing practices of hospitality and tourism firms to identify how these ideas or activities affect the enhancement of innovation performance of the said sector. Secondly, this research provides evidence that business ethics diffusion is predominantly important in the innovation performance process. Business ethics diffusion forms collaborative environment that facilitates interaction and adds to commitment among workers for knowledge, information, and ideas sharing. Thirdly, our study results bolster knowledge sharing's moderating function in the connection between business-ethics diffusion and innovation performance. Moreover, outcomes declare that knowledge-sharing is the fundamental factor that boosts the impacts of business-ethics diffusion on innovation performance. In other words, the success of the innovation performance is mostly dependent on the willingness of employees to design, integrate, and share information and knowledge. Fourth, we suggest a research model that shows that service innovation act as a bridge between business ethics diffusion and innovation performance. This study contributes to an improved understanding of business ethics diffusion on innovation performance through means of mediated service innovation.

*5.2. Practical Contributions*

This research's practical findings offer an enhanced understanding of the fundamental relationship between business-ethics diffusion, service innovation, and knowledge-sharing and the innovation performance of hospitality and tourism firms. This research has great

value for the management of hospitality and tourism firms as a reference for knowledge-sharing, fostering service innovation, and achieving innovation performance in its firms. The current study contributes to prior knowledge in the following ways. First, this research suggests that scholars and practitioners must focus on processes that encourage innovation practices because innovation performance is critical for hospitality and tourism firms to survive, grow, and develop in an environment of high market complexity and increasing competitive intensity. Nevertheless, prior studies are not sufficient to guide hospitality and tourism firms to improve their innovation performance. That is why this study provides hospitality and tourism firms with vital practical guidance and deeper consideration of the association between business ethics diffusion, service innovation, knowledge sharing, and innovation performance. Furthermore, results show that both business ethics diffusion and service innovation are more crucially linked with innovation performance compared to knowledge sharing. More specifically, it is clear that service innovation is driver of innovation performance through the predominant role of business-ethics diffusion for hospitality and tourism firms. Accordingly, the management of hospitality and tourism firms should focus on structuring ideas that enhance service innovation and motivate employees to take part in the process of innovation performance. Second, our results support that knowledge-sharing is a stimulating factor in facilitating business-ethics diffusion and increasing innovation performance. This support can enhance the association between business-ethics diffusion and innovation performance. At last, through investigating the impact of control variables, for instance, firm type, size, etc; the results revealed that a firm's size is critically linked to innovation performance. This means that firms with more resources and funds have further opportunities to recycle and innovate their products.

*5.3. Limitations and Future Directions*

This study provides some good knowledge and understanding of prior literature, but it also has some limitations. First, the quantitative research design is used for data collection; in the future, cross-sectional research or qualitative methods could be used. Second, business ethics diffusion is suggested as having a vital impact on the success of innovation performance; future studies should examine the influence of other variables on service innovation that stimulates high innovation performance. Third, we selected hospitality and tourism firms to implement this beneficial empirical model, but in the future, other sectors, such as manufacturing, distribution, energy firms, etc., should be selected. Finally, we considered service innovation as a mediating variable and knowledge-sharing as a moderating variable; other variables may be used as a moderator or mediators in this model. In addition, in the future, such studies can be conducted in Western culture to generalize the results of the current study.

**Author Contributions:** Conceptualization, H.Y.; methodology, M.S.; software, H.Y.; validation, J.Z.; formal analysis, J.Z.; investigation, J.R.-S.; resources, A.J.; data curation, A.J.; writing, J.Z.; draft preparation, H.Y.; writing—review and editing, M.S.; supervision and project administration, Z.Y. Authors are willing to publish this manuscript. All authors have read and agreed to the published version of the manuscript.

**Funding:** No external funding was received.

**Institutional Review Board Statement:** The study was conducted in accordance with the Declaration of Helsinki and approved by the Institutional Review Board (or Ethics Committee) of Govt Commerce College Mansehra (23100 code GMAN-134-56) for studies involving humans.

**Informed Consent Statement:** Informed consent was obtained from all subjects involved in the study.

**Data Availability Statement:** Not applicable.

**Conflicts of Interest:** The authors declare no conflict of interest.

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
