# Peer review of "Towards Innovation Performance of the Hospitality and Tourism Industry: Interplay among Business Ethics Diffusion, Service Innovation, and Knowledge-Sharing"

_sustainability, doi:10.3390/su15010886_

Round 1

Reviewer 1 Report

Greetings,

The paper is very well done. Smaller corrections need to be made to make the paper even better. Before selection 2.1. Business Ethics Diffusion and Innovation Performance you need to write a paragraph about these variables. Explain why a reverse Likert scale was used where the best grade had the lowest grade and vice versa. It must be presented how many companies are covered by Business

Age, Business Size and other characteristics of companies. Tables 3 and 4 should be better presented in the results. The rest of the paper is well written, nothing needs to be corrected.

All the best.

Author Response

Dear Editor (Sustainability)

Manuscript ID Sustainability 2071455

The paper titled " Towards Innovation Performance of Hospitality and Tourism Industry: Interplay among Business Ethics Diffusion, Service Innovation and Knowledge Sharing" has been reviewed and improved as per comments of the referee(s). These changes are incorporated and highlighted in the text; the details of the changes are as follows:

                        Reviewer 1

Suggestion Incorporated

The paper is very well done. Smaller corrections need to be made to make the paper even better.

Before selection 2.1. Business Ethics Diffusion and Innovation Performance you need to write a paragraph about these variables.

Explain why a reverse Likert scale was used where the best grade had the lowest grade and vice versa.

It must be presented how many companies are covered by Business Age, Business Size and other characteristics of companies. Tables 3 and 4 should be better presented in the results. The rest of the paper is well written, nothing needs to be corrected.

All the best.

Thanks for your appreciation.

One paragraph on each variable is added before 2.1.

It is typing mistake we used a similar Likert scale pattern where the best grades the had highest grades and vice versa.

Table 1 is explained

Thanks for your appreciation.

Thanks.

Reviewer 2 Report

The authors have undertaken a significant and current research problem, which is the combination of business ethics with innovative activities in service companies, in this case, in hotels.
The article is written in a model way. A theoretical introduction was made, and a review of the basic but the latest literature, research questions and research hypotheses were put forward. The hypotheses were verified using appropriate statistical tools (SEM). Appropriate conclusions were drawn based on the conducted research.
The research sample is sufficient. The authors also indicated research limitations and the possibility of conducting further research using other variables and service or industrial sectors.
An interesting research result demonstrates the great importance of business ethics in innovative activities. It should be noted here that the study was conducted in Pakistan, a country with a different culture, religion and approach to doing business than the countries of Western Europe or North America. An interesting comparison would be to carry out such research in European or American hotels. It is known that in different cultures (and in the same countries), there are different approaches to business ethics. The use of an ethical attitude often results from fashion, the desire for additional profit, and market requirements, but there is no more profound inner conviction in the managers themselves.
It should also be noted that it is the countries of Southeast Asia are considered to be the markets with the cheapest labour force, and workers there are forced to work for the lowest wages and exploited by employers. How does such an unethical attitude relate to the results of this study? Do different ethical perspectives govern the hotel industry?
One should also pay attention to the concept of innovation and innovativeness. In services, particularly in the hotel industry, it is more difficult to introduce them than in the industry. Contacting the service recipient with the service provider is necessary, which limits innovative activities but enhances ethical attitudes. In industry, however, the manufacturer usually remains anonymous, which does not force him to adopt a rigorous ethical approach.

Author Response

Dear Editor (Sustainability)

Manuscript ID Sustainability 2071455

The paper titled " Towards Innovation Performance of Hospitality and Tourism Industry: Interplay among Business Ethics Diffusion, Service Innovation and Knowledge Sharing" has been reviewed and improved as per comments of the referee(s). These changes are incorporated and highlighted in the text; the details of the changes are as follows:

Reviewer 2 (Changes are shown in Green color)

S. No

Comments

Authors Response

1

The authors have undertaken a significant and current research problem, which is the combination of business ethics with innovative activities in service companies, in this case, in hotels.
The article is written in a model way. A theoretical introduction was made, and a review of the basic but the latest literature, research questions and research hypotheses were put forward. The hypotheses were verified using appropriate statistical tools (SEM). Appropriate conclusions were drawn based on the conducted research.

Thank you for supporting our work and way of writing. It is nice to read some positive comments from the reviewer.

2

The research sample is sufficient. The authors also indicated research limitations and the possibility of conducting further research using other variables and service or industrial sectors.
An interesting research result demonstrates the great importance of business ethics in innovative activities. It should be noted here that the study was conducted in Pakistan, a country with a different culture, religion and approach to doing business than the countries of Western Europe or North America. An interesting comparison would be to carry out such research in European or American hotels. It is known that in different cultures (and in the same countries), there are different approaches to business ethics. The use of an ethical attitude often results from fashion, the desire for additional profit, and market requirements, but there is no more profound inner conviction in the managers themselves.

Thank you for some positive comments and appreciation. Furthermore, we have added recommendations to conduct cross-cultural research using the same model and scale, in the recommendation section (page 13, lines 403-405).

3

It should also be noted that it is the countries of Southeast Asia are considered to be the markets with the cheapest labour force, and workers there are forced to work for the lowest wages and exploited by employers. How does such an unethical attitude relate to the results of this study? Do different ethical perspectives govern the hotel industry?

Cheap labor is a competitive edge for firms operating in Pakistan (the country where the study was conducted). This matter was kept in mind as some of the studies also pointed out the unethical attitude toward employees

(like see

1.      Zafar, R., & Lodhi, S. (2015). The Study Of Ethical Issues In Restaurant Of Karachi, Pakistan. International Journal of Scientific & Technology Research, 4 (11), 370374.

2.      Sarwar, H., Ishaq, M. I., Amin, A., & Ahmed, R. (2020). Ethical leadership, work engagement, employees’ well-being, and performance: a cross-cultural comparison. Journal of Sustainable Tourism28(12), 2008-2026.

3.      Fenitra, R. M., Abbas, A., Ekowati, D., &Suhairidi, F. (2022). Strategic Intent and Strategic Leadership: A Review Perspective for Post-COVID-19 Tourism and Hospitality Industry Recovery. The Emerald handbook of destination recovery in tourism and hospitality, 23-44.). Before starting the work on this paper, this matter was considered, however it was not mentioned under the ethical considerations head. Detail is added on page 5, lines 197-202.

4

One should also pay attention to the concept of innovation and innovativeness. In services, particularly in the hotel industry, it is more difficult to introduce them than in the industry. Contacting the service recipient with the service provider is necessary, which limits innovative activities but enhances ethical attitudes. In industry, however, the manufacturer usually remains anonymous, which does not force him to adopt a rigorous ethical approach.

The reviewer has pointed out a very good point that ethics may limit the innovativeness of a firm. However, this was not in the scope of current research, in future research such matters can be evaluated.

Reviewer 3 Report

The manuscript entitled "Towards Innovation Performance of Hospitality and Tourism Industry: Interplay among Business Ethics Diffusion, Service Innovation and Knowledge Sharing" is relevant to the journal's aims and scope. However, it still has some issues that need to be addressed. Below is the list of my suggestions for manuscript enhancement.

- I suggest that the authors, in addition to the specified indicators of fit indices, also show the Tucker–Lewis index (TLI) and standard root mean square residual (SRMR).

- Authors should provide a reference indicating Cronbach's alpha coefficient threshold value of >0.6 (line 242).

- In the notes, authors should explain the meaning of the abbreviations from the first line of the table in Table 1.

- Before presenting the results of the CFA and SEM analyses, the authors should have conducted a pilot study to test and validate the proposed model and confirm the factor structure using exploratory factor analysis (EFA).

How and where was the questionnaire tested and eventually adjusted to the expected sample?

- There are numerous scientific studies on the topic covered by the authors of this study. The authors must offer new research and, within the discussion, compare the results of their sample to those of other samples. Twenty-six of the 42 references are older than five years, which is not enough when writing about a topic that has been written about by many authors.

Author Response

Dear Editor (Sustainability)

Manuscript ID Sustainability 2071455

The paper titled " Towards Innovation Performance of Hospitality and Tourism Industry: Interplay among Business Ethics Diffusion, Service Innovation and Knowledge Sharing" has been reviewed and improved as per comments of the referee(s). These changes are incorporated and highlighted in the text; the details of the changes are as follows:

Reviewer 3 (Changes are shown in blue color)

S. No

Comments

Authors Response

1

The manuscript entitled "Towards Innovation Performance of Hospitality and Tourism Industry: Interplay among Business Ethics Diffusion, Service Innovation and Knowledge Sharing" is relevant to the journal's aims and scope. However, it still has some issues that need to be addressed. Below is the list of my suggestions for manuscript enhancement.

Thank you for supporting our work and way of writing.

2

- I suggest that the authors, in addition to the specified indicators of fit indices, also show the Tucker–Lewis index (TLI) and standard root mean square residual (SRMR).

Values of TLI and SRMR has been added on page 8, line 291 and 292.

3

- Authors should provide a reference indicating Cronbach's alpha coefficient threshold value of >0.6 (line 242).

Reference is added.

4

- In the notes, authors should explain the meaning of the abbreviations from the first line of the table in Table 1.

Abbreviations are explained.

5

- Before presenting the results of the CFA and SEM analyses, the authors should have conducted a pilot study to test and validate the proposed model and confirm the factor structure using exploratory factor analysis (EFA).

Explanation added. Please check the highlighted section.

6

How and where was the questionnaire tested and eventually adjusted to the expected sample?

A questionnaire was tested through pilot study and with CFA.

7

- There are numerous scientific studies on the topic covered by the authors of this study. The authors must offer new research and, within the discussion, compare the results of their sample to those of other samples. Twenty-six of the 42 references are older than five years, which is not enough when writing about a topic that has been written about by many authors.

More references were added.

Reviewer 4 Report

  Dear authors,

First of all, I am glad to have the opportunity to read the article entitled “Towards Innovation Performance of Hospitality and Tourism Industry: Interplay among Business Ethics Diffusion, Service Innovation and Knowledge Sharing”, that I have read with great interest.

The paper has some points that need the author’s attention as below

Please justify choosing the study context:   Hospitality and Tourism Industry, as no mention of the uniqueness of that context in the manuscript

In the measurement section, please do not make a subsection for each measure, it looks like writing a book, not scientific research. Moreover, Whenever you are referring to other authors like created by, introduced by, adopted from -- please add author before or instead of  numbers [XX] 

My main concern is that you employed CFA for reliability and validity but was surprised that you did not employee SEM for the structural model and hypotheses justification. Please justify or employ SEM.

There is a disconnect between the research question and the research methodology. In other words, the research method you adopted can be applied to other fields as well, such as changing hotel management to restaurant management. Therefore, it is recommended that you further explain the problem and research methods here.

The Theoretical part and Discussion can be enriched with more literature. Such as 

Elshaer, I.A.; Azazz, A.M.S. Amid the COVID-19 Pandemic, Unethical Behavior in the Name of the Company: The Role of Job Insecurity, Job Embeddedness, and Turnover Intention. Int. J. Environ. Res. Public Health 202219, 247. https://doi.org/10.3390/ijerph19010247

Elshaer, I.A.; Azazz, A.M.S.; Mahmoud, S.W.; Ghanem, M. Perceived Risk of Job Instability and Unethical Organizational Behaviour Amid the COVID-19 Pandemic: The Role of Family Financial Pressure and Distributive Injustice in the Tourism Industry. Int. J. Environ. Res. Public Health 202219, 2886. https://doi.org/10.3390/ijerph19052886

Author Response

Dear Editor (Sustainability)

Manuscript ID Sustainability 2071455

The paper titled " Towards Innovation Performance of Hospitality and Tourism Industry: Interplay among Business Ethics Diffusion, Service Innovation and Knowledge Sharing" has been reviewed and improved as per comments of the referee(s). These changes are incorporated and highlighted in the text; the details of the changes are as follows:

Reviewer 4 (Changes are shown in pink color)

S. No

Comments

Authors Response

1

First of all, I am glad to have the opportunity to read the article entitled “Towards Innovation Performance of Hospitality and Tourism Industry: Interplay among Business Ethics Diffusion, Service Innovation and Knowledge Sharing”, that I have read with great interest.

Thank you. It is nice to read some positive comments from the reviewer.

2

The paper has some points that need the author’s attention as below

Please justify choosing the study context:   Hospitality and Tourism Industry, as no mention of the uniqueness of that context in the manuscript

The explanation is added. Please see the highlighted section in the introduction.

3

In the measurement section, please do not make a subsection for each measure, it looks like writing a book, not scientific research. Moreover, Whenever you are referring to other authors like created by, introduced by, adopted from -- please add author before or instead of  numbers [XX] 

Suggestions have been incorporated.

4

My main concern is that you employed CFA for reliability and validity but was surprised that you did not employee SEM for the structural model and hypotheses justification. Please justify or employ SEM.

Our study used CFA for establishing validity and reliability, but SEM was not used for evaluating other direct or indirect relationships. There is some strong literature support involving the use of CFA and other techniques. Some examples from the recent past are given below:

Ali, Z., Gongbing, B., &Mehreen, A. (2018). Does supply chain finance improve SMEs performance? The moderating role of trade digitization. Business Process Management Journal. 26(1), 150-167.

Aloulou, W. J. (2016). Predicting entrepreneurial intentions of final year Saudi university business students by applying the theory of planned behavior. Journal of Small Business and Enterprise Development. 23(4), 1142-1164.

Turel, O., & Gil-Or, O. (2018). Neuroticism magnifies the detrimental association between social media addiction symptoms and wellbeing in women, but not in men: a three-way moderation model. Psychiatric Quarterly89(3), 605-619.

Stoermer, S., Hitotsuyanagi-Hansel, A., & Froese, F. J. (2019). Racial harassment and job satisfaction in South Africa: The moderating effects of career orientations and managerial rank. The International Journal of Human Resource Management30(3), 385-404.

Yasir, M., & Majid, A. (2019). Boundary integration and innovative work behavior among nursing staff. European Journal of Innovation Management. https://doi.org/10.1108/EJIM-02-2018-0035

Banerjee, P., Gupta, R., & Bates, R. (2017). Influence of organizational learning culture on knowledge worker’s motivation to transfer training: testing moderating effects of learning transfer climate. Current Psychology36(3), 606-617.

Following the flow of these published papers, we also used more than one technique to evaluate the relationships.

5

There is a disconnect between the research question and the research methodology. In other words, the research method you adopted can be applied to other fields as well, such as changing hotel management to restaurant management. Therefore, it is recommended that you further explain the problem and research methods here.

As per recommendation of reviewer, text has been changed.

6

The Theoretical part and Discussion can be enriched with more literature. Such as 

Elshaer, I.A.; Azazz, A.M.S. Amid the COVID-19 Pandemic, Unethical Behavior in the Name of the Company: The Role of Job Insecurity, Job Embeddedness, and Turnover Intention. Int. J. Environ. Res. Public Health 202219, 247. https://doi.org/10.3390/ijerph19010247

Elshaer, I.A.; Azazz, A.M.S.; Mahmoud, S.W.; Ghanem, M. Perceived Risk of Job Instability and Unethical Organizational Behaviour Amid the COVID-19 Pandemic: The Role of Family Financial Pressure and Distributive Injustice in the Tourism Industry. Int. J. Environ. Res. Public Health 202219, 2886.https://doi.org/10.3390/ijerph19052886

Thanks for suggesting such valuable research papers.

Reviewer 5 Report

The language is so full of mistakes that it makes it impossible to understand what the paper is about. The abstract is promising an interesting story, but further on the text is completely incomprehensive, so any sensible review is impossible.

I suggest to the writers to find a qualified translator before resubmitting the text.

Author Response

Dear Editor (Sustainability)

Manuscript ID Sustainability 2071455

The paper titled " Towards Innovation Performance of Hospitality and Tourism Industry: Interplay among Business Ethics Diffusion, Service Innovation and Knowledge Sharing" has been reviewed and improved as per comments of the referee(s). These changes are incorporated and highlighted in the text; the details of the changes are as follows:

Reviewer 5 (Changes are shown in Grey color)

Comments

Authors Response

The language is so full of mistakes that it makes it impossible to understand what the paper is about.

The abstract is promising an interesting story, but further on the text is completely incomprehensive, so any sensible review is impossible.

All the text has been proofread and corrections are made throughout the text.

Abstract revised

I suggest to the writers to find a qualified translator before resubmitting the text.

Thanks for the suggestion.

Round 2

Reviewer 3 Report

I would like to thank the authors for taking the reviewer's suggestions into account. The proposed changes and additions to the paper have been accepted, so in my opinion, the paper is accepted for publication in the scientific journal Sustainability.

Author Response

Dear Editor,

Sustainability

Reference to Manuscript-ID: 2071455, Title: Towards Innovation Performance of Hospitality and Tourism Industry: Interplay among Business Ethics Diffusion, Service Innovation and Knowledge Sharing.

 As per the suggestions of reviewer 5, all the changes are incorporated by the authors. A detail of comments and responses is given below.

Reviewer 3

Comments

Authors Response

I would like to thank the authors for taking the reviewer's suggestions into account. The proposed changes and additions to the paper have been accepted, so in my opinion, the paper is accepted for publication in the scientific journal Sustainability.

Thank you very much for accepting paper

Reviewer 4 Report

can accept in its current form 

Author Response

Dear Editor,

Sustainability

Reference to Manuscript-ID: 2071455, Title: Towards Innovation Performance of Hospitality and Tourism Industry: Interplay among Business Ethics Diffusion, Service Innovation and Knowledge Sharing.

 As per the suggestions of reviewer 5, all the changes are incorporated by the authors. A detail of comments and responses is given below.

Reviewer 4

Comments

Authors Response

can accept in its current form 

Thank you very much for accepting paper

Reviewer 5 Report

While the language is improved, there are still numerous mistakes and incomprehensive sentences. I am attaching a text where I market just some of the cases I noticed when reading- I did not mark the entire text, since I'm not the language editor.

From the content point of view, I lack more information on the survey. You only present the statistics, but I believe more details on the survey, including the questions and answers obtained, from which you draw the relevant conclusions should be presented as well. This would increase the credibility of the results.  

Author Response

Dear Editor,

Sustainability

Reference to Manuscript-ID: 2071455, Title: Towards Innovation Performance of Hospitality and Tourism Industry: Interplay among Business Ethics Diffusion, Service Innovation and Knowledge Sharing.

As per the suggestions of reviewer 5, all the changes are incorporated by the authors. A detail of comments and responses is given below.

Reviewer 5 (Changes are shown in Grey color)

Comments

Authors Response

Proofread the whole document,

All the text has been proofread and corrections are made throughout the text (highlighted in gray color)

Provide more information about the survey conducted and present results.

The details of survey conduct is already provide in Methodology section and highlighted. Results are also presented.

Round 3

Reviewer 5 Report

The authors have improved the quality of language significantly. There are still some typos to be corrected, in particular the spacing. I would still like to learn more about the questionnaire (type of questions, response rates to specific questions, etc.) which are not evident from the model, but if other reviewers find the paper ok, I will not insist anymore.